# Nicotine Dependence in Patients with Major Depressive Disorder and Psychotic Disorders and Its Relationship with Quality of Life

**DOI:** 10.3390/ijerph182413035

**Published:** 2021-12-10

**Authors:** Peizhi Wang, Edimansyah Abdin, P.V. Asharani, Vanessa Seet, Fiona Devi, Kumarasan Roystonn, Ying Ying Lee, Laxman Cetty, Wen Lin Teh, Swapna Verma, Yee Ming Mok, Mythily Subramaniam

**Affiliations:** 1Research Division, Institute of Mental Health, Buangkok Green Medical Park, 10 Bunagkok View, Singapore 539747, Singapore; Edimansyah_Abdin@imh.com.sg (E.A.); Asharani_PEZHUMMOOTTIL_VASUDEVAN_N@imh.com.sg (P.V.A.); ai_ling_seet@imh.com.sg (V.S.); Fiona_Devi_Siva_Kumar@imh.com.sg (F.D.); k_roystonn@imh.com.sg (K.R.); Ying_Ying_LEE@imh.com.sg (Y.Y.L.); Laxman_Cetty@imh.com.sg (L.C.); Wen_Lin_TEH@imh.com.sg (W.L.T.); Mythily@imh.com.sg (M.S.); 2Early Psychosis Intervention Programme, Institute of Mental Health, Singapore 539747, Singapore; Swapna_Verma@imh.com.sg; 3Office of Education, Duke-NUS Medical School, 8 College Road, Singapore 169857, Singapore; 4Department of Mood and Anxiety, Institute of Mental Health, Buangkok Green Medical Park, 10 Buangkok View, Singapore 539747, Singapore; Yee_ming_mok@imh.com.sg

**Keywords:** nicotine dependence, quality of life, major depressive disorder, psychotic disorder

## Abstract

The aim of the current study was to examine the associations between nicotine dependence and quality of life (QOL) among individuals diagnosed with major depressive disorder (MDD) or psychotic disorders. **Methods:** A total of 378 participants diagnosed with either MDD or psychotic disorders were recruited. The Fagerstorm Test for Nicotine Dependence was used to measure the level of nicotine dependence. The SF-12 health survey questionnaire was used to measure the QOL. **Results:** The prevalence of nicotine dependence was 23.3% in this sample population. For those diagnosed with MDD, moderate level of nicotine dependence was negatively associated with Vitality and Mental Component Score. For those diagnosed with a psychotic disorder high nicotine dependence was negatively associated with Role Emotional, Mental Health and Mental Component Score. **Discussion:** Compared to the general population, the prevalence of smoking in this psychiatric population was 2.4 times higher, while that of nicotine dependence was seven times higher. Individuals with psychotic disorder generally reported better QOL as compared to individuals with MDD. QOL differed across diagnostic groups with regards to socio-demographics, such as age, ethnicity, marital status, education, employment status and monthly income. Among individuals with MDD and psychotic disorders, different levels of nicotine dependence resulted in different levels of association with QOL. More research is needed to better understand the differences in QOL among the varying levels of nicotine dependence.

## 1. Introduction

Smoking is a leading cause of preventable death worldwide [1]. The World Health Organisation (WHO) has estimated that every year more than eight million people die due to smoking-related diseases [1]. Smoking-related diseases contribute significantly to the global rise in non-communicable diseases in both developed and developing countries. Smoking has been linked to cancer, cardiovascular diseases, stroke, and other diseases [2]. Moreover, the mortality rate of smokers is three times higher, and they die an average of 10 years earlier than non-smokers [3]. Cigarettes are the most commonly used form of tobacco product worldwide [1]. Other forms of tobacco use include cigars, roll your own tobacco, pipes, shisha, and the most recently introduced electronic nicotine delivery systems (ENDS) [1]. Nicotine is the key chemical compound that causes tobacco use disorder. In recent years, the design and contents of emerging tobacco products have made smoking novel, highly addictive, and more damaging to health as compared to the past [1]. For example, the introduction of ENDS is appealing because of its slim design, different colours and flavours [4]. Furthermore, the formulation of e-cigarettes increases the rate of nicotine delivery and decreases the harsh sensation in the throat, making it appealing and addictive [4].

A steady decline in smoking prevalence rates in western countries has been reported [5]. Similarly, in Singapore, the prevalence of daily smoking had dropped from 14.3% in 2010 to 12% in 2017 [6,7]. An epidemiological study conducted in 2012 among individuals aged 18 years and above showed that 16.3% of the resident population were current smokers [8]. More recently, a national survey using the same methodology showed no change in the prevalence of smoking in Singapore (16.1%) [9]. The authors reported that the prevalence of nicotine dependence was on a downward trend from 4.5% in 2012 to 3.3% in 2016 [9].

Notwithstanding the above, there are certain subgroups who are more prone to smoking and nicotine dependence. Individuals with mental disorders smoke at rates approximately double that compared to individuals who do not have mental disorders [10], especially those diagnosed with major depressive disorder (MDD) and schizophrenia [11]. The predisposition towards smoking could be due to poor coping strategies, smoking as a self-medication to cope with symptoms of mental illness, or as social reinforcement where smoking is a social activity or culture in mental health facilities and to avoid the higher severity of withdrawal symptoms [12,13]. They also tend to smoke more heavily, which means that this group faces greater risks of health problems related to smoking [10]. In addition to this, individuals with mental disorders experience up to 25 years of reduced life expectancy compared to the general population [14]. Much of this reduction in life expectancy can be attributed to smoking-related diseases, making it important to track and monitor smoking and nicotine dependence rates among the mentally ill [11].

Both schizophrenia and depression can affect an individual’s quality of life. Socio-demographic correlates such as female gender, younger age, marital status, and lower education are associated with better quality of life (QOL) in schizophrenia patients [15]. Positive symptoms such as hallucinations and delusions in schizophrenia can significantly impair functioning [16]. Negative symptoms are similarly associated with poorer mental and physical functioning [16]. Studies have indicated that MDD accounts for severely impaired work ability, social ability, and emotional and physical functioning. Those with MDD are less likely to graduate from school, more likely to report poor quality of interpersonal relationships, and more likely to divorce than individuals without MDD [17].

WHO defined quality of life as an individual’s perception of their position in life in the context of the culture and value systems in which they live and in relation to their goals, expectations, standards, and concerns [18]. Many QOL measurement tools have been developed and validated over the years. These tools assess QOL as a multidimensional construct, including physical, psychological, and social functioning [19]. With the emphasis on holistic care, QOL has gained traction in mental health services to address the psychosocial aspect of mental health. A systematic review conducted in 2019 concluded that QOL is an important endpoint in medical and health research [20].

Studies have investigated the relationship between smoking cessation and QOL. In 2014, a systematic review concluded that smoking cessation improves QOL and mood while reducing symptoms of anxiety and depression [21]. Although effective, smoking cessation is often overshadowed by managing an individual’s mental disorder coupled with the difficulty of getting smokers to quit and to stay quit [22]. Despite the well-established risk of diseases associated with smoking, to the best of our knowledge, there is a dearth of studies investigating associations between nicotine dependence and QOL among individuals with mental illness. Studies have also not compared nicotine dependence across mental disorders and whether it impacts QOL differently across disorders. The association between QOL and smoking may have an influence on the ability of the smoker to reduce or stop smoking. Goldenberg and colleagues [23] suggested, that understanding the relationship between QOL and smoking initiation can inform prevention efforts. By understanding the impact of nicotine dependence on QOL, the information can be used to improve the motivation to quit, and enhance relapse prevention strategies. 

Therefore, the present study aims to examine the associations between nicotine dependence and QOL among individuals diagnosed with MDD or psychotic disorders. We will also examine how varying levels of nicotine dependence affect QOL. Such findings are necessary to better inform interventions aimed at smoking cessation and improving the physical and psychological health of people with serious mental illness.

## 2. Methodology

### Participants

Data were collected from October 2018 to February 2020. Participants were recruited from the inpatient wards or outpatient clinics at the Institute of Mental Health, the only tertiary psychiatric care hospital in Singapore. Participants were recruited through (a) referrals by the attending clinician and (b) self-referrals in response to the recruitment posters displayed at the clinics. Trained study team members screened the participants to ensure that the inclusion criteria were met. Participants had to meet the following eligibility criteria: (1) Singapore citizen or permanent resident (2) aged 21 to 65 years old (3) diagnosis of either MDD or schizophrenia spectrum and other psychotic disorders. The study team members confirmed the eligibility of the participants against the medical records before administering the questionnaires. The study excluded participants who did not have the mental capacity to consent to the study or did not meet the diagnostic criteria. The diagnosis of mental disorders was based on the fourth edition of the Diagnostic and Statistical Manual of Mental Disorders (DSM-IV-TR) criteria [24]. The survey was administered in either one of the four local languages: English, Chinese, Malay, or Tamil, as preferred by the participants. Ethics approval was obtained from the Institutional Research Review Committee, and Domain Specific Review Board of the National Healthcare Group (Ref: 2018/00772), and written consent was granted by all participants.

## 3. Materials Used

### 3.1. Socio-Demographic Questionnaire

A modified version of the Global Adult Tobacco Survey [25] was used to capture data that included sociodemographic information (e.g., age, sex, ethnicity, education, housing, and income) and questions regarding smoking status. The classification of a smoker, non-smoker, and past smoker was based on the definitions from the National Health Interview Survey [26]. Participants who self-reported a lifetime smoking of at least 100 cigarettes and were smoking at the time of the survey were classified as smokers, and those who had smoked 100 cigarettes in their lifetime and had quit smoking were classified as former smokers [26]. Non-smokers were those who had never smoked 100 cigarettes in their lifetime. For the purpose of data analysis and interpretation, ex-smokers were subsumed under the category for current smokers to form a new category of “ever-smokers” because of the small sample size and the need to keep to the convention set by the first article published for this project [27]. The diagnosis was based on the clinician report, which was extracted from the medical records.

### 3.2. Fagerstrom Test of Nicotine Dependence (FTND)

The level of nicotine dependence was measured using the Fagerstrom Test for Nicotine Dependence (FTND) [28]. This scale consists of six items regarding daily cigarette consumption and assesses the extent of cigarette smoking and smoking behaviour. The number of response options varies for each item, and response options are given a score of 0, 1, 2, or 3. Items are then summed to give a total score ranging from 0 to 10. Scores of 0 to 4 are classified as low dependence, 5 to 7 as moderate and 8 and above are classified as high dependence [27]. We categorized those with scores of 5 and above as having nicotine dependence, as defined by a previous study [9]. The FTND has been shown to have adequate validity [28] and is also a reliable tool for the assessment of smoking behaviour in patients with schizophrenia [29].

### 3.3. 12-Item Short Form Survey (SF-12)

The SF-12 questionnaire developed for the Medical Outcomes Study was used to assess QOL [30]. This scale includes 12 items covering eight domains of health status: Physical Functioning (PF), Role-Physical (RP), Bodily Pain (BP), General Health (GH), Vitality (VT), Social Functioning (SF), Role-Emotional (RE) and Mental Health (MH). The Physical Component Score (PCS) summarizes PF, RP, BP, and GH while the Mental Component Score (MCS) reflects VT, SF, RE, and MH. These scores range from 0 to 100 and are calculated by the standard algorithm described in the SF-12 manual [25]. A score of 0 indicates the lowest level of health measured and a score of 100 indicates the highest level of health. The SF-12 has been tested and validated in Asian populations, including Hong Kong [31], and Singapore [32].

## 4. Statistical Analysis

Descriptive statistics were tabulated for the overall sample. Means and standard deviations were calculated for continuous variables and frequencies and percentages for categorical variables. Multiple linear regression was used to examine the associations between QOL, socio-demographic variables, and mental disorders in the overall sample. Diagnosis-specific analyses were also conducted to explore the socio-demographic correlates of QOL as well as the association between nicotine dependence and QOL. The following variables were controlled for: age, gender, ethnicity, education, marital status, employment status, monthly income, and type of diagnosis. Respondents who had any missing values on any variables were removed from the analysis: one current smoker and one ex-smoker had missing values on the FTND questionnaire. All statistical analyses were carried out using Statistical Analysis Software (SAS) System version 9.2. Statistical significance was evaluated at the ≤0.05 level using two-sided tests.

## 5. Results

### 5.1. Descriptive Statistics

A total of 380 participants were recruited, and two cases were removed listwise from the analysis due to missing data. Table 1 shows the sociodemographic characteristics and smoking habits of the participants. Among the 378 participants recruited, about half the participants (46.3%) were diagnosed with MDD, while the remaining 53.7% were diagnosed with psychotic disorders. About 66.3% of participants with MDD were between 21 and 40 years old, while 35% of the participants with a psychotic disorder were 21 to 40 years old. The prevalence of smoking was 39.5%, as reported in an earlier paper [27]. Among the sample of MDD individuals, 46.9% were non-smokers, 36.6% had low dependence, 14.3% had moderate, and 2.3% had a high nicotine dependence. Among the sample of psychotic disorders, 56.2% were non-smokers, 14.8% had low dependence, 19.2% had moderate, and 9.9% had a high nicotine dependence.

### 5.2. Association of Sociodemographic Factors with QOL

Table 2 shows the association of sociodemographic factors with QOL for the MDD sample. Participants aged 41 to 65 years old were negatively associated with PCS (β = −5.01, *p* = 0.007) and positively associated with MCS (β = 6.33, *p* = 0.009). Compared to Chinese ethnicity, Indians were negatively associated with BP (β =−6.48, *p* = 0.02), PCS (β = −5.03, *p* = 0.009). Being single was negatively associated with BP (β = −6.24, *p* = 0.024), VT (β = −4.55, *p*=0.033) and PCS (β = −4.25, *p* = 0.025). Those with secondary educations were negatively associated with BP (β = −7.04, *p* = 0.045) and MH (β =−6.55, *p* = 0.025), as compared to those who were university educated. Those with pre-university were negatively associated with VT (β = −5.26, *p* = 0.011) and MCS (β = −5.70, *p* = 0.018). Compared to those with a monthly income of below $2000, those with a monthly income of $2000 to $3999 were negatively associated with VT(β = −4.42, *p* = 0.039).

Among those with a psychotic disorder (Table 3), 41 to 65 years was negatively associated with PCS (β = −3.59, *p* = 0.013). Compared to Chinese ethnicity, Malays were positively associated with VT (β = 6.10, *p* = 0.016), SF (β = 4.90, *p* = 0.048) and MCS (β = 5.21, *p* = 0.024). Indians were also positively associated with SF (β = 8.89, *p* = 0.007) and MCS (β = 7.29, *p* = 0.017). Compared to those with a university education, those who were vocationally trained were negatively associated with RP (β = −5.81, *p* = 0.045) and VT (β = −8.41, *p* = 0.016). Those who are economically inactive were negatively associated with SF (β = −6.97, *p* = 0.032) as compared to those who were employed.

### 5.3. Association between Nicotine Dependence and QOL in MDD and Psychotic Disorder

As reflected in Table 4, moderate nicotine dependence was negatively associated with VT (β = −5.11, *p* = 0.041) and MCS (β = −5.77, *p* = 0.046) in the MDD subgroup. High nicotine dependence was not significantly associated with any QOL subdomains in the MDD. Among the psychotic disorder subgroup, high nicotine dependence was negatively associated with RE (β = −7.81, *p* = 0.018), MH (β = −6.59, *p* = 0.021) and MCS (β = −7.09, *p* = 0.012). 

## 6. Discussion

Our findings showed that the prevalence of current smoking (39.4%) in the psychiatric population was 2.4 times higher than that of the general population (16.1%) [9]. This is in line with previous studies where a higher prevalence of smoking was reported among people with mental illness [10,33]. We also found that 23.3% of the psychiatric population was nicotine dependent. This figure is seven times higher than that of the general population, which is reported to be at 3.3% [9]. Similarly, this is consistent with an overseas study whereby those with psychiatric disorders were two to 16 times more likely to have nicotine dependence than those without these diagnoses [34]. Researchers had also reported that those with a diagnosed mental illness were 25% less likely to quit smoking [10,33] and had a lower readiness to quit [27]. The high prevalence of nicotine dependence and lower probability of smoking cessation in this group is a concern for public health professionals and physicians to note, as they would require more intensive intervention to reduce or stop their smoking behavior.

Both psychosis and MDD can affect an individual’s overall quality of life. In this study, QOL differed across the two diagnostic groups based on sociodemographics factors such as age, ethnicity, marital status, education, employment status, and monthly income. Older individuals with MDD reported better mental health scores (MCS) and poorer physical health scores (PCS) than their younger counterparts. The association of older age with poorer PCS was not unusual, as older participants would more likely experience limitations to their physical health as compared to those in the younger age group, leading to lower PCS. Previous studies have suggested that a curvilinear relationship between age and QOL exists, whereby life satisfaction peaked at 65 years, and happiness peaked at 50 years [35,36]. This may explain why compared to the younger counterparts, the older individuals had a higher scores in their MCS. Compared to those of Chinese ethnicity, Indians were negatively associated with BP, which may have negatively affected the PCS. The differences of pain perception within the ethnic groups are similar to what has been previously reported in the general population [37]. In a national survey of Singaporean adults, Indian participants reported greater pain severity when compared with both Malay and Chinese participants [37].

Similarly, being single and having a secondary school education was negatively associated with BP. These relationships were not observed among individuals with psychosis. Being married enables individuals to share resources such as finances, food, social support, workload and responsibilities. These resources may act as a buffer against BP and VT, which are not readily available to single people. Individuals who have attained an education lower than university were negatively associated with various subdomains of QOL. Generally, individuals with higher education can hold a better-paying job which could, in turn, lead to better health and well-being [38]. They might also be more knowledgeable about the determinants of their health and are hence able to adopt a better lifestyle, leading to better functional outcomes. 

As compared to those with MDD, individuals diagnosed with psychotic disorder reported better QOL in almost all subdomains (refer to Table 3). This result is in line with previous studies whereby individuals with schizophrenia reported better QOL as compared to those with MDD [39,40]. The authors suggest that there are underlying differences, as individuals with psychosis may lack insight into their illness and social environment. Thus, they may develop protective strategies (i.e., denial, minimization) and assign meaning to their lives, leading to better QOL than individuals with depression [40]. On the other hand, individuals with depression experience persistent feelings of sadness, hopelessness, decreased motivation, energy and anhedonia. These may in turn affect their functioning, and they are therefore more inclined to report poorer QOL [39]. 

Among those with psychosis, Malays were positively associated with VT, SF and MCS, while Indians were positively associated with SF and MCS as compared to Chinese. One possible explanation may be their cultural value system, as these are unique to population groups [41,42]. Similar to individuals with MDD, those with an education lower than university were negatively associated with QOL (RP and VT)., while those who were economically inactive were negatively associated with SF. We are not able to explain this association. 

In terms of nicotine dependence among the MDD sample, moderate levels of nicotine dependence were negatively associated with VT. However, this association was not observed in the psychosis sample, while no significant association was observed for the high nicotine dependence group. This could be due to the analysis being underpowered, as only four participants in the MDD group demonstrated high nicotine dependence. It has been reported that smokers experience feelings of fatigue [43]. It may be possible that the low moods might have exacerbated the feelings of fatigue in the MDD sample or the fatigue may be due to smoking-related disease. Despite most smokers wanting to quit [29], many perceive that smoking provides them with mental health benefits. Smokers describe smoking as being able to alleviate emotional problems such as feelings of depression and anxiety, reduce the effect of stigma, stabilise mood, and for relaxation as well as relieving stress [44,45]. However, there is evidence indicating otherwise. A meta-analysis reported that smokers have poorer mental health as compared to those who have quit [21]. The authors found that quitting was associated with a decrease in anxiety and depression symptoms and improvement in their psychological quality of life [21]. They concluded that should individuals with mental disorders cease smoking, it is unlikely to exacerbate their symptoms and might be therapeutic instead [21].

Among the psychosis subgroup, individuals with high levels of nicotine dependence had poorer mental QOL in domain RE, MH and MCS. Individuals with psychosis typically experience discrimination in society [46]. They are more likely to live in a state of hardship partly due to the disorder itself, as it can lead to loss of employment, housing, and conflict with family members [46]. This stigma of being shunned by society can lead to social marginalization, discrimination in access to and quality of health services, and impoverishment, thus affecting QOL. To mitigate these, individuals may smoke heavily to manage mood, have a sense of control, self-medicate, and be socially accepted [44]. A study looking at community living psychiatric clients described their most comforting times being when their nurses spent time smoking with them. They felt that during this time they were less judgmental and were connected to them as a whole person and not a fragmented being [44]. Another study looking at homelessness reported a sense of camaraderie among smokers who used smoking as a way to initiate social interaction with others [47].

There were some limitations in this study. Firstly, our study was cross-sectional in nature; there was no way of determining causality in the association between smoking and QOL. Secondly, we did not collect information on physical illnesses that may affect their QOL. Thirdly, we did not assess the participant’s illness severity, which may be related to the degree of nicotine dependence. Lastly, we used face-to-face interviews, and participants may not have reported accurate cigarette consumption due to social desirability. This is especially true for females, where there is a social expectation that they should not smoke, as it is inappropriate or unfeminine [48]. On the other hand, males may over-report their cigarette consumption to portray a more “cool” or “tough” image.

Given the limitations, future research could expand the scope of the current study by looking at how cutting down on cigarettes and smoking cessation over multiple timepoints affects QOL. Nevertheless, the study adds insight into the smoking behaviour and QOL among the psychiatric population in Singapore and provides preliminary information that will aid in the planning of smoking cessation programs. 

## 7. Conclusions

The prevalence of nicotine dependence among the psychiatric population was seven times higher than the general population. Although this figure is much higher than the general population, it is in line with the international prevalence among the psychiatric population. Individuals with psychotic disorder generally reported better QOL scores across all subdomains as compared to those with MDD. Furthermore varying levels of nicotine dependence affected those diagnosed with MDD and psychosis differently. The reasons behind the higher prevalence of smoking and higher nicotine dependence among the psychiatric population is multi-faceted, ranging from the need self-medication to gaining social acceptance. Despite the perceieved benefits of managing symptoms of anxiety and bonding with other smokers in their community, studies suggested that in the long-term, smoking results in poorer mental and physical health, especially among the psychiatric patients [21].

The high rates of smoking and nicotine dependence among the psychiatric population are worrying, as smoking greatly increases the risk of other health issues. Public health professionals can be more aggressive in intervening to reduce nicotine consumption as the first step. Health care providers, especially in mental health institutions, may wish to periodically assess their clients’ smoking patterns and behaviours. Gradual reduction of smoking or harm reduction may be introduced to those who are undecided on quitting, while programs that assist with quitting may be introduced to those who are motivated. Costs associated with smoking cessation, such as nicotine replacement therapy, could be subsidized to ensure affordability. Finally, some psychoeducation on the long-term effects of smoking could be introduced to this population to help them understand the insidious effects of long-term heavy smoking. Helping them identify healthier ways of coping with their symptoms and gaining social acceptance could go a long way to improve the physical, mental and social wellbeing of this population.

## Figures and Tables

**Table 1 ijerph-18-13035-t001:** Socio-demographic characteristics of psychiatric patients.

	MDD (n = 175)		Psychotic Disorder (n = 203)	
	n	%	n	%
Overall				
Age				
21–40	166	66.3	71	35.0
41–65	59	33.7	132	65.0
Gender				
Male	90	51.4	120	59.1
Female	85	33.7	83	40.9
Ethnicity				
Chinese	117	66.9	160	78.8
Malay	28	16.0	27	13.3
Indian	26	14.9	13	6.4
Others	4	23	3	1.5
Educational level				
Primary school	17	9.7	34	16.7
Secondary school	28	16.0	74	36.5
Pre-university (JC/diploma)	67	38.3	43	21.2
Vocational Institute/ITE	26	14.9	27	13.3
University	37	21.1	25	12.3
Marital status				
Married	33	18.9	29	14.3
Single	115	65.7	147	72.4
Separated/divorced/widowed	27	15.4	27	13.3
Employment status				
Employed	84	48.0	94	46.3
Unemployed	71	40.6	95	46.8
Economically inactive	20	11.4	14	6.9
Monthly Income				
Below SGD $2000	129	73.7	177	87.2
SDG $2000 to SGD $3999	35	20.0	19	9.4
SGD $4000 and above	11	6.3	7	3.4
Smoking Status				
Non-Smoker	82	46.9	114	56.2
Ever-smoker	93	53.1	89	43.8
FTND				
Non-smoker	82	46.9	114	56.2
Low (0–4)	64	36.6	30	14.8
Moderate (5–7)	25	14.3	39	19.2
High (8 and above)	4	2.3	20	9.9

MDD only includes unipolar depression, those with bipolar disorders were excluded.

**Table 2 ijerph-18-13035-t002:** Association between socio-demographic with QOL in the MDD sample (n = 175).

	Physical				Mental				Summary	
	PF (β)	RP (β)	BP (β)	GH (β)	VT (β)	SF (β)	RE (β)	MH (β)	PCS (β)	MCS (β)
**Age (Ref: 21–40)**										
41–65	−3.74	−1.78	−3.51	0.14	0.86	3.25	4.45	3.92	**−5.01**	**6.33**
**Gender (Ref: Male)**										
Female	−1.27	−2.35	0.51	0.44	−2.84	−0.54	−1.93	0.04	−0.92	−1.05
**Ethnicity (Ref: Chinese)**										
Malay	1.03	−0.20	0.59	−1.72	−1.00	−2.61	−2.59	−3.79	1.45	−4.16
Indian	−1.55	−3.24	**−6.48**	−3.87	0.99	−4.49	0.14	1.13	**−5.03**	1.07
Others	−1.86	−2.88	1.81	6.21	3.8246	2.28	−1.91	−1.32	2.65	−0.32
**Marital status (Ref: Married)**										
Single	−3.06	−1.72	**−6.24**	−0.74	**−4.55**	0.2	0.12	0.12	**−4.25**	0.63
Separated/divorced/widowed	−2.64	−0.14	−4.30	−0.99	−2.79	−0.80	0.55	1.48	−3.28	1.14
**Educational level (Ref: University)**										
Primary school	−1.58	−1.25	−4.56	−3.15	−0.65	2.5	1.42	−1.59	−3.30	1.37
Secondary school	−1.49	−2.90	**−7.04**	−3.19	−5.14	−0.81	−1.54	**−6.55**	−3.09	−3.84
Pre-university (JC/diploma)	1.32	0.87	0.11	−3.06	**−5.26**	−1.29	−3.91	−4.22	1.68	**−5.70**
Vocational Institute/ITE	−0.70	−3.05	−5.14	1.96	−0.07	−1.24	−1.49	−1.03	−1.99	−0.49
**Employment status (Ref: Employed)**										
Unemployed	−0.75	−2.94	−1.14	1.08	−2.43	−1.21	−3.85	−2.19	−0.28	−3.02
Economically inactive	−1.06	0.99	3.18	−1.64	−4.27	−0.8	0.45	1.23	0.02	−0.61
**Monthly Income (Ref: Below SGD $2000)**										
SDG $2000 to SGD $3999	2.18	−0.21	1.9	−1.22	**−4.42**	1.16	0.13	−1.09	1.23	−1.86
SGD $4000 and above	−0.12	−1.96	−4.4	−2.2	−2.23	0.17	−0.09	1.63	−3.1	0.99

MDD only included unipolar depression, those with bipolar disorder were excluded. Linear Regression (controlled for socio-demographic factors and nicotine dependence). PF: Physical functioning; RP: Role limitations due to physical problems; BP: Bodily pain; GH: General health; VT: Vitality; SF: Social functioning; RE: Role limitations due to emotional problems; MH: Mental health; PCS: Physical Component Score; MCS: Mental Component Score.

**Table 3 ijerph-18-13035-t003:** Association between socio-demographic with QOL in the psychotic sample (n = 203).

	Physical				Mental				Summary	
	PF (β)	RP (β)	BP (β)	GH (β)	VT (β)	SF (β)	RE (β)	MH (β)	PCS (β)	MCS (β)
**Age (Ref: 21–40)**										
41–65	−2.37	−2.18	−3.14	−0.90	−0.43	−0.56	1.81	1.1	**−3.59**	2.2
**Gender (Ref: Male)**										
Female	−0.24	1.19	−1.22	−0.53	0.34	3.01	2.04	0.21	−0.66	1.91
**Ethnicity (Ref: Chinese)**										
Malay	−1.16	0.36	−3.81	0.34	**6.1**	**4.9**	0.63	3.3	−2.20	**5.21**
Indian	−2.15	3.54	−2.51	3.36	5.94	**8.89**	4.97	3.26	0.4	**7.29**
Others	−1.60	−1.45	−3.62	3.4	0.22	−4.77	1.06	1.45	−1.97	0.78
**Marital status (Ref: Married)**										
Single	−0.62	0.57	3.04	0.85	0.74	2.53	−1.47	−1.76	1.79	−0.88
Separated/divorced/widowed	−3.18	−0.72	0.68	3.02	−0.27	0.44	−1.70	1.32	−0.61	0.67
**Educational level (Ref: University)**										
Primary school	−0.99	−0.97	−4.89	−5.39	−2.92	0.86	1.62	−3.36	−3.32	−0.48
Secondary school	−3.19	−1.61	−0.76	−3.13	−0.51	−0.82	−1.49	−2.66	−2.07	−1.22
Pre-university (JC/diploma)	−2.70	−2.74	−0.83	−1.33	−0.62	1.44	−1.23	−1.38	−2.19	0.1
Vocational Institute/ITE	−2.80	**−5.81**	−2.88	−1.21	**−8.41**	−3.62	−5.57	−3.37	−2.84	−5.35
**Employment status (Ref: Employed)**										
Unemployed	−1.04	−1.61	−2.95	−1.13	−2.00	−1.53	−2.61	−1.46	−1.45	−1.98
Economically inactive	0.07	0.47	−2.94	0.98	−4.31	**−6.97**	−2.58	2.03	0.38	−4.80
**Monthly Income (Ref: Below SGD $2000)**										
SDG $2000 to SGD $3999	2.08	0.36	1.29	3.34	−0.10	1.34	0.98	−0.22	1.87	−0.68
SGD $4000 and above	2.17	3.85	1.53	4.69	1.71	3.64	6.91	2.99	1.97	4.64

MDD only included unipolar depression, bipolar disorders were excluded. Linear Regression (controlled for socio-demographics factors and nicotine dependence). PF: Physical functioning; RP: Role limitations due to physical problems; BP: Bodily pain; GH: General health; VT: Vitality; SF: Social functioning; RE: Role limitations due to emotional problems; MH: Mental health; PCS: Physical Component Score; MCS: Mental Component Score.

**Table 4 ijerph-18-13035-t004:** Association of Nicotine Dependence with QOL among those with MDD (n = 175) and Psychotic Disorder (n = 203).

	**Physical**				**Mental**				**Summary**	
	**PF**	**RP**	**BP**	**GH**	**VT**	**SF**	**RE**	**MH**	**PCS**	**MCS**
**MDD**										
FTND (Ref. Non-smoker)										
Low (0–4)	0.15	1.38	2.09	0.92	0.35	−0.00	−1.43	−2.39	2.26	−2.11
Moderate (5–7)	−1.28	−0.39	0.81	0.44	**−5.11**	−1.16	−4.70	−5.12	1.58	**−5.77**
High (8 and above)	0.3	−9.27	6.2	5.43	−6.54	−6.43	−9.90	0.1	1.94	−7.12
	**Physical**				**Mental**				**Summary**	
	**PF**	**RP**	**BP**	**GH**	**VT**	**SF**	**RE**	**MH**	**PCS**	**MCS**
**Psychotic Disorder**										
FTND (Ref. Non-smoker)										
Low (0–4)	4.1	3.82	−0.21	1.63	−0.49	−1.15	1.97	0.84	2.85	−0.58
Moderate (5–7)	3.02	4.02	2.8	0.59	2.1	4.73	2.29	−0.46	3.42	1.06
High (8 and above)	−0.29	−3.66	0.05	0.9	−2.15	−1.80	**−7.81**	**−6.59**	1.74	**−7.09**

MDD only includes unipolar depression, those with bipolar disorders were excluded. Linear Regression (controlled for socio-demographics factors). PF: Physical functioning; RP: Role limitations due to physical problems; BP: Bodily pain; GH: General health; VT: Vitality; SF: Social functioning; RE: Role limitations due to emotional problems; MH: Mental health; PCS: Physical Component Score; MCS: Mental Component Score.

## Data Availability

The data presented in this study are available on request from the corresponding author. The data are not publicly available due to institutional policies.

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
