# Peer review of "Nicotine Dependence in Patients with Major Depressive Disorder and Psychotic Disorders and Its Relationship with Quality of Life"

_ijerph, 2021, doi:10.3390/ijerph182413035_

Round 1

Reviewer 1 Report

Thanks for the author for addressing my comments.

Reviewer 2 Report

I have no further comment.

This manuscript is a resubmission of an earlier submission. The following is a list of the peer review reports and author responses from that submission.

Round 1

Reviewer 1 Report

Review for \Nicotine Dependence in Patients
with Major Depressive Disorder and Psychotic
Disorders and its Relationship with Quality of
Life" (Submission to International Journal of
Environmental Research and Public Health)
My recommendation for this paper is rejection. First, I would like to thanks
for the author provide a statistcs analysis and interpret the result correctly.
However, the analysis in this paper is too basic. It just looks like a class assign-ment. Please consider doing the analysis extensively. Also, this paper lack of inovative and scienti c sounds. I suggest the research design and methods can be extend.The statistical analysis is very basic. However, I do thanks for the author gave a correct interpretation.

Reviewer 2 Report

Re: ijerph-1389953, Nicotine dependence in patients with major depressive disorder and psychotic disorders and its relationship with quality of life

  1. The rationale to compare the patients with depression and psychotic disorder should be explained more clearly since these two disorders are very different.
  2. Since this study is cross-sectional designed, the influence from poor mental health to smoking (self-medication hypothesis) should be addressed in the introduction. Furthermore, the potential implication that enhances the mental health may be helpful from nicotine control, should be mentioned in the discussion.
  3. Table 3 and table 4 could be merged into one table. Stratified analysis should also conducted in Table 1, and 2.

Reviewer 3 Report

I found the study very interesting and well writen in general. I have little to add, excepting two aspects:

  • I missed a control group to better understand the nature of the differences found between MDD and Psychotic Disorders groups. I know that the target population is the clinic one, but I believe that a control group (without any disorders) could give a deepen view of how other variables (i.e. nicotine dependence) are present in the general population, comparing to both groups. I suggest to include this point in future proposals.
  • Some writing mistakes have been detected in the text and I suggest to review language aspects.

I would like to congratulate authors.
